# Radiation Dose Management in Computed Tomography: Introduction to the Practice at a Single Facility

Yusuke Inoue 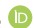

Department of Diagnostic Radiology, Kitasato University School of Medicine, Sagamihara 252-0374, Japan; inoueys@kitasato-u.ac.jp

**Abstract:** Although the clinical benefits of computed tomography (CT) are undoubtedly high, radiation doses received by patients are also relatively high; therefore, radiation dose management is mandatory to optimize CT radiation doses and prevent excessive radiation events. This article describes CT dose management practice at a single facility. Many imaging protocols are used in CT depending on the clinical indications, scan region, and CT scanner; thus, managing the protocols is the first step for optimization. The appropriateness of the radiation dose for each protocol and scanner is verified, while answering whether the dose is the minimum to obtain diagnostic-quality images. Moreover, examinations with exceptionally high doses are identified, and the cause and clinical validity of the high dose are assessed. Daily imaging practice should follow standardized procedures, avoiding operator-dependent errors, and information required for radiation dose management should be recorded at each examination. The imaging protocols and procedures are reviewed for continuous improvement based on regular dose analysis and multidisciplinary team collaboration. The participation of many staff members in the dose management process is expected to contribute to promoting radiation safety through increased staff awareness.

**Keywords:** computed tomography; excessive dose event; imaging protocol; radiation dose management





## 1. Introduction

Computed tomography (CT) offers detailed morphological information, and additionally functional information with the aid of contrast media, in a short examination period. Although CT is established as an indispensable imaging modality in modern medicine, it delivers relatively high radiation doses to patients, and an increased risk of cancer is of significant concern [1–4]. Therefore, optimization of CT radiation dose is of vital importance, and radiation dose management is required at each imaging facility.

Here, CT radiation dose management performed at our facility, a university hospital in Japan, is described. We have six CT scanners provided by four manufacturers: among them, one scanner is situated in the emergency department and another in the operation department. We perform about 45,000 CT examinations annually. Radiologists, radiological technologists, and nurses engage in CT practice; unfortunately, no medical physicists are involved although the critical roles of medical physicists are undisputed [5,6]. In Japan, most medical physicists dedicate their efforts to radiotherapy.

## 2. Management of Imaging Protocols

### 2.1. Significance of Protocol Management

In interventional radiology, optimizing the radiation dose is challenging because the dose in a given procedure depends on various factors such as patient size, disease complexity, vascular anatomy, and operator's technique, in addition to the imaging parameters for fluoroscopy and radiography. In contrast, radiation dose in CT is mainly determined by the imaging protocol, CT scanner, and patient size and is more controllable and predictable. Establishing appropriate imaging protocols and applying an optimal protocol for each

examination are effective for optimizing patient dose. Reducing the radiation dose tends to increase image noise; excessive reduction may impair diagnostic performance. For optimization of radiological imaging, the radiation dose should be reduced while maintaining acceptable image quality and satisfactory diagnostic performance. When an increase in image noise is judged to be acceptable based on the evaluation of image quality, revision of the imaging protocol is considered to reduce radiation exposure. The use of diagnostic reference levels (DRLs) is recommended for optimizing radiological imaging [7]. The DRL is determined based on dose surveys in a country or region and is usually defined as the 75th percentile of the dose distribution for standard-sized patients. If the radiation dose used in a facility is higher than the DRL, it should trigger inspection for dose reduction.

### 2.2. Preparation of Subdivided Imaging Protocols

In some facilities, a small number of typical imaging protocols are prepared, and an operator changes the imaging parameters on the CT operation console according to the requirement of each examination. This practice system reduces the burden of protocol management; however, it depends on the operator whether appropriate imaging is performed. In our facility, we subdivide imaging protocols and prepare many protocols in advance, according to the clinical demands. An operator applies an optimal protocol to each examination and follows the protocol with no need for further customization. Although this system increases the burden of protocol management, it reduces the operator's burden and error during each examination. Moreover, it contributes to the internal standardization of CT practice in our facility and allows for continuous improvement through detailed radiation dose management. We evaluate radiation dose and image quality for each protocol and monitor the frequency of utilization of each protocol and the validity of the protocol selection for each examination.

For brain CT without contrast enhancement, for example, we have prepared various protocols, including ones for acute cerebral infarction, hydrocephalus, and children, in addition to the routine protocol. When a patient is referred for brain CT, the optimal protocol is selected from the predefined protocols. To detect acute cerebral infarction, we increase radiation exposure and consequently decrease image noise to facilitate the recognition of subtle changes in image appearance. Patients with hydrocephalus may repeatedly undergo CT scans to reveal changes in ventricular size, resulting in a high cumulative dose. Because high-quality images are not required to assess the ventricular size, we the reduce radiation exposure and simultaneously change the setting for hybrid reconstruction (image reconstruction involving both traditional filtered back-projection and iterative reconstruction) to depress an increase in image noise due to the dose reduction. Moreover, the imaging planes are set to avoid exposing the orbits to radiation, and images parallel to the orbitomeatal line are created by reformatting. For children, we reduce the radiation dose with special care while maintaining clinically acceptable image quality, considering the high radiosensitivity and long life expectancy of children [8].

### 2.3. Preparation of the Protocol List

We prepared a protocol list using general-purpose word-processing software to facilitate the appropriate use and review of the imaging protocols. In the protocol list, all imaging protocols are listed, and the items in Table 1 are described for each protocol. In our facility, the protocol name is registered in Japanese in the radiology information system (RIS), while it is in English in the CT scanner. The protocol number is common to both the RIS and CT scanners, which facilitates comparison between the two systems. In the protocol list, items the operator should pay special attention to during examination, i.e., the scan range, the amount and infusion duration of contrast media, and the start timing of imaging after contrast infusion, are displayed in red. In addition, a scan range list was prepared as an adjunct to the protocol list. Whereas the scan ranges are verbally explained in the protocol list, in the scan range list, the upper and lower margins of the ranges are displayed on localizer images to aid the operators' visual recognition.

**Table 1.** Items described for each protocol in the protocol list.

- Protocol name and protocol number in the RIS
- Protocol name on the CT console
- Corresponding order item in the ordering system
- Scan range (for each imaging phase in multiphase imaging)
- Amount, concentration, and infusion duration of contrast media
- Start timing of imaging after contrast infusion
- Images to be reconstructed and reformatted
- Remarks at pre-check
- Clinical indications

*2.4. Tabulation of Imaging Parameters*

To supplement the imaging protocol list, we prepared the imaging parameter list using general-purpose spreadsheet software. Detailed parameters of each imaging protocol for image acquisition and reconstruction are described in the imaging parameter list (Table 2). These parameters are registered in the CT scanners; however, the parameters of different protocols cannot be displayed simultaneously for comparison. The external parameter table facilitates comparison among CT scanners and protocols for review purposes. When modifying the parameters, modification is usually required for different scanners, as well as for different but related protocols. First, a revised version of the imaging parameter list is proposed, and the responsible person and other CT team members confirm the validity of the revision. Then, the new parameters are registered to the CT scanners with the aid of the revised list. This modification procedure helps avoid errors in determining and registering the parameters.

**Table 2.** Items described for each protocol in the imaging parameter table.

- Scan mode (axial or helical)
- Tube voltage
- Method to determine tube current (fixed or AEC-based)
- AEC parameter settings, which vary by manufacturer
- Beam width
- Rotation time
- Pitch factor
- Slice thickness/slice increment
- Reconstruction kernel
- Parameters for hybrid reconstruction (related to noise reduction effects)
- Scan field of view/display field of view

## 3. Overview of CT Radiation Dose Management

*3.1. Dose Indices Used for Radiation Dose Management*

Radiation dose indices such as the volume CT dose index (CTDIvol) and the dose length product (DLP) are automatically calculated by CT scanners and used for radiation dose management [9]. CTDIvol is an index of the intensity of radiation exposure at the imaging section. It is calculated based on the absorbed dose measured on a 16 or 32 cm dosimetry phantom and the imaging parameters. DLP is the integral of CTDIvol over the entire scan range and represents the total irradiation exposure in an imaging series. When multiple series of images are acquired during one examination, such as for dynamic CT, the total DLP for the examination is also calculated, in addition to the CTDIvol and DLP for each series.

DLP can be converted to an effective dose, an indicator of the risk of stochastic effects, i.e., carcinogenic and hereditary effects, by multiplying the DLP by a conversion factor provided by the International Commission on Radiological Protection (ICRP) [9]. Because the primary concern with CT irradiation is an increased risk of cancer [1–4], the total DLP in an examination is considered to represent the potential detriment of an examination.

Thus, we primarily use the total DLP in an examination for radiation dose management and refer to the DLP and CTDIvol in each series when necessary.

In pediatric body CT, the CT scanner may use a 16 or 32 cm phantom to calculate CTDIvol and DLP, depending on the scanner type and patient age. When using a 16 cm phantom for calculation, the value of CTDIvol is approximately double than when using a 32 cm phantom. We convert the values based on a 32-cm phantom to those based on a 16 cm phantom by multiplying by a conversion factor of approximately 2.

The absorbed dose varies depending on the section size, even when radiation output from the CT scanner is identical. Size-specific dose estimate (SSDE) is a relatively new index of CT radiation dose and is calculated considering the effect of the section size [10,11]. Whereas CTDIvol represents radiation output from the scanner, SSDE is considered a better indicator of the patient's absorbed dose than CTDIvol. However, the estimation of SSDE requires the section size, and a CT scanner does not automatically provide the SSDE value. CTDIvol and DLP are still typically used for CT radiation dose management.

### 3.2. Automatic Exposure Control

Severe attenuation of X-rays in the imaging section decreases the detected X-rays at the same radiation exposure, resulting in increased image noise. In larger patients, radiation exposure should be increased to prevent the loss of diagnostic performance due to an excessive increase in noise. The X-ray output is proportional to the tube current of the CT scanner, and radiation exposure is manipulated mainly by adjusting the tube current. We use automatic exposure control (AEC) for most of our CT examinations and adjust the radiation exposure according to body size. The AEC system evaluates X-ray attenuation for each patient and each location mainly using localizer images and automatically modulates the tube current [12–14]. AEC contributes to dose reduction by achieving appropriate adjustment of irradiation according to the degree of X-ray attenuation. Although many facilities do not use this function for pediatric brain CT [15,16], the head size of children increases rapidly with growth, especially in the early postnatal period; AEC is recommended for appropriate dose modulation according to the head size [17]. In addition, automatic tube voltage selection techniques aid the selection of an appropriate tube voltage for each imaging series to reduce the radiation dose depending on the patient size and the type of examination (plain CT, contrast-enhanced CT, or CT angiography) [18,19].

### 3.3. The Need to Consider Body Size in Radiation Dose Management

When using AEC, DLP varies with body size and correlates well with body weight [20–22]. Because a higher DLP in a larger patient does not imply an excessive radiation dose beyond that required, body size should be considered in radiation dose management. Usually, DRLs are established only using dose indices for standard-sized patients, and each facility compares their doses to the DRLs only for standard-sized patients [7]. We determine median doses for standard-sized patients and compare them to the Japanese DRLs [23] every year and early after changing parameters. However, the relationship between body size and dose varies with the CT scanner and imaging protocol [20,21], and it is desirable to evaluate doses for various body sizes. DRLs for different body sizes have been reported [24].

### 3.4. Radiation Dose Management Considering Body Size

We analyze the radiation dose for each imaging protocol and each CT scanner to determine the standard dose according to body weight [20]. Total DLP is plotted against weight using our clinical data, and a linear regression equation is determined and used to estimate the standard DLP according to weight (Figure 1).

Visual evaluation of the plots is also important because a straight line may not be suitable for fitting of the plots due to technical or patient factors. For example, maximum and minimum tube currents may be set in AEC software. Tube currents above the maximum current are not applied, even if the body is very large. Reaching a plateau at heavy weights in a DLP–weight relationship suggests the application of the maximum current in large

patients and the possibility of insufficient exposure and a consequent increase in image noise. In this case, CT images acquired in large patients are reviewed to determine whether increased noise is clinically acceptable. In contrast, if the dose decrease with decreasing weight is unclear in small patients on the DLP–weight plots, the possibility that the required dose is exceeded due to a high minimum tube current setting should be considered. On the other hand, because head growth precedes body growth early after birth, the radiation dose in pediatric brain CT shows biphasic changes. In younger, smaller patients, the dose increases with age and weight is more pronounced, while in older, larger patients, the dose increase is slower. Thus, we determine two linear regression equations separately for younger and older children [25].

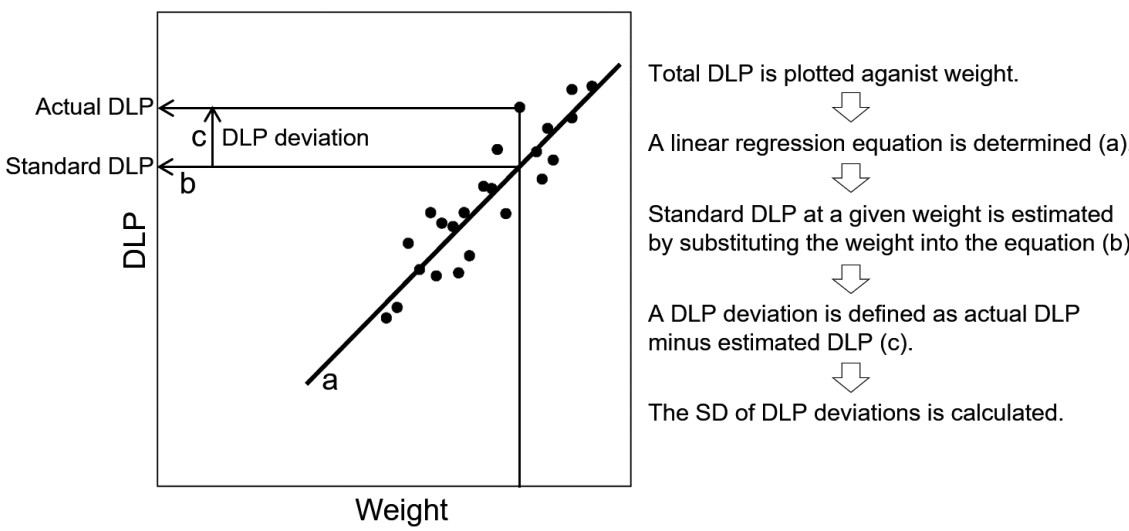

**Figure 1.** Evaluation of the DLP–weight relationship for each protocol. A small number of dummy data are shown for explanation purposes.

### 3.5. Standard Dose Tables for Radiation Dose Management

We calculate standard DLP values at body weights of 40, 60, and 80 kg using standard dose determination equations derived from the dose analysis and create standard dose tables for radiation dose management [20]. Using these tables, radiation doses are compared between CT scanners and imaging protocols to identify unreasonable inconsistencies. When a potential problem is found, we first check the imaging parameters registered to the CT scanner to ensure that there are no input errors. Then, imaging parameters are compared between scanners and protocols on the imaging parameter list, which may reveal inappropriate settings. When there is a significant difference in standard DLP values between two scanners, the images are reviewed, and the possibility of reducing the dose delivered by the higher-dose scanner is considered. Instead, increasing the dose on the lower-dose scanner may be considered.

### 3.6. Identifying Examinations Delivering Exceptionally High Doses

In routine radiation dose management, examinations delivering exceptionally high radiation doses are identified, and the cause of the high dose is investigated. Depending on the cause, a measure to prevent recurrence may be planned and undertaken.

We systematically identify outliers in the DLP–weight relationship [20]. When analyzing DLP–weight plots to obtain standard dose determination equations, we calculate DLP deviations, defined as actual DLP values provided by the scanner minus DLP values estimated based on the weight and equation, and determine their standard deviations (SDs) (Figure 1). In routine radiation dose management, the weight for each examination is substituted into the standard dose determination equation to estimate the DLP at that weight (Figure 2). The DLP deviation for each examination is calculated and divided by the

predefined SD to calculate the Z-value. Large Z-values indicate exceptionally high doses, and such examinations are subject to investigation. Exceptionally high doses often have valid reasons and do not necessarily mean excessive doses; nevertheless, this process aids the detection of examinations with excessive doses. A fixed threshold DLP value may be used, regardless of body size, to identify potential excessive-dose examinations; however, this method will mostly extract examinations performed in large patients. Instead, outliers may be searched for by visual inspection of dose-weight plots, but this process is subjective and time-consuming. Setting a Z-value threshold allows outliers to be found efficiently based on fixed criteria.

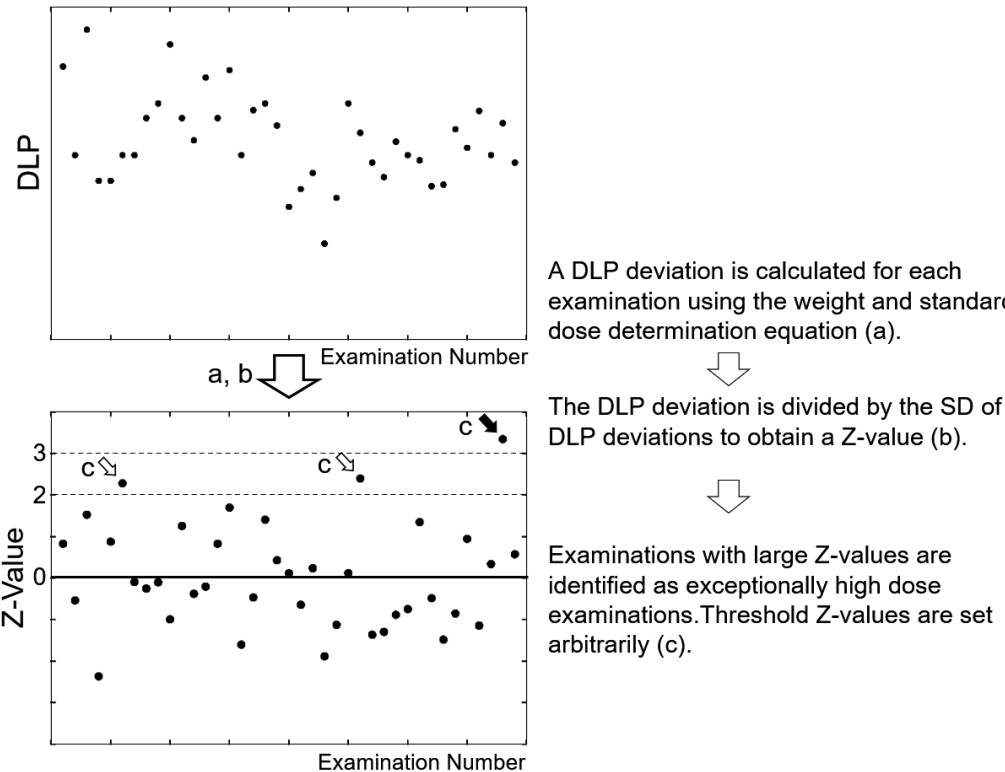

A DLP deviation is calculated for each examination using the weight and standard dose determination equation (a).

The DLP deviation is divided by the SD of DLP deviations to obtain a Z-value (b).

Examinations with large Z-values are identified as exceptionally high dose examinations. Threshold Z-values are set arbitrarily (c).

**Figure 2.** Identification of examinations delivering exceptionally high doses. Dummy data are used for presentation.

### 3.7. Dose Reference Tables for Referring Physicians

The standard dose tables are used for radiation dose management in the radiology department. We also prepare dose reference tables for referring physicians and make them accessible via the electronic medical system terminal used to order examinations. The detriments of irradiation to patients depend on the radiation dose, which varies among the order items and facilities. Dose information for each order item at each facility is necessary for the referring physician to consider the indication for the examination, select an order item, and explain the examination to the patient. Since DLP is unfamiliar to referring physicians and patients, we provide a standard effective dose value for each order item in the dose reference table. The dose for each order item is analyzed to determine the linear regression equation for the DLP–weight relationship. Using this equation, the standard DLP at a body weight of 60 kg is calculated and multiplied by the ICRP conversion factor to obtain the standard effective dose [9]. The standard effective dose represents the effective dose estimated at 60 kg for the respective order item.

There are some problems to be noted in converting DLP to an effective dose. The ICRP conversion factors for pediatric body CT are applicable to the DLP calculated based on a 16- cm phantom. When the DLP provided by the CT scanner is based on a 32 cm phantom, it should be converted to one based on a 16-cm phantom before multiplication

by the ICRP conversion factor. In positron emission tomography (PET)/CT, CT images from the head to the proximal thigh or foot are acquired in an imaging series, and the scanner provides a single DLP value. The DLP value may be multiplied by the conversion factor for the body trunk, which leads to an overestimation of the effective dose due to inaccurate estimation for the head, neck, and lower extremities [26–28]. In CT angiography and CT venography of the lower extremities, although irradiation to the lower extremities contributes significantly to DLP, the contribution to the effective dose is limited because of the low radiosensitivity of the lower extremities. Application of the conversion factor for the trunk leads to a severe overestimation of the effective dose [29,30].

## 4. Imaging Procedures Based on and toward Radiation Dose Management

### 4.1. Steps for Good CT Practice

For high-quality, safe radiological imaging services, imaging procedures and imaging protocols should be standardized in each facility. All staff should be familiarized with the standardized imaging procedures via documentation and training and comply with the procedures during clinical practice. In addition, dose-related records should be made during each examination to support efficient and effective radiation dose management. Efforts are made to continuously improve imaging procedures and protocols based on the results of dose management.

### 4.2. Request for CT by the Referring Physician

When requesting a CT examination, the referring physician selects an order item using the electronic ordering system and enters clinical information and the purpose of the examination. At our facility, various order items are set according to the imaging region, imaging phase, and clinical indication. The order item buttons are colored according to the radiation dose so that the referring physician is aware of the dose level. Furthermore, the referring physician can view the dose reference table on the terminal used for ordering and recognize the standard effective dose for the order item. When the referring physician selects an order item, a pop-up window appears presenting the following text: "Please consider the indication, range, and frequency, taking account of the possibility of radiation-induced cancer". After fixing the order, a document of explanation for the patient is printed out, which includes a general explanation of radiation exposure in CT and effective dose values for natural radiation and certain radiological examinations. The referring physician explains the dose, justification, and optimization of the individual examination with reference to this document and the dose reference table presenting a standard effective dose value for each order item.

### 4.3. Preparation on the Day before the Examination

On the day before the examination, CT orders are checked in the radiology department to confirm the clinical indications and determine detailed imaging procedures. This process is called a pre-check. The pre-checks are performed by radiologists and radiological technologists using the RIS and by referring to the protocol list. To optimize the radiation dose, the scan range and the number of scan phases are minimized according to the purpose of the examination. To aid the selection of an appropriate imaging protocol, imaging protocol options for each order item are presented at the top of the protocol list, along with notes for protocol selection. During the pre-check, the radiology department staff display the CT order on the RIS terminal and select the optimal imaging protocol based on the requested order item, clinical information, examination purpose, previous examinations, etc. The name and number of the selected protocol are entered in the pre-check comment field in the RIS, along with any special remarks if necessary. Special remarks include changes in the scan range, the addition or omission of a scan phase, consideration of a cardiac device, and specification of the scanner to be used. When a contrast study is planned in a patient with impaired renal function, the comment "20% reduction of contrast media, 100-kV imaging" is entered. An asterisk is placed at the beginning of a special remark to

prevent the operator from missing the remark during the examination. Standardized text is provided for frequently used remarks.

*4.4. Operation on the RIS during the Examination*

In the morning, before the start of patient imaging, the protocol list and scan range list are opened on the RIS terminal located next to the CT operation console. When performing each examination, the operator displays information about the examination stored in the RIS, which includes pre-check comments, and also checks the description in the protocol list and scan range list for the protocol indicated in the pre-check. The operator enters the body height and weight of the patient into the RIS and transfers the examination information in the RIS, including height and weight, to the CT scanner. Weight is essential for radiation dose management and is measured in the CT division before the examination.

*4.5. Operation on the CT Console during the Examination*

The operator selects the imaging protocol on the CT console following the description in the pre-check field of the RIS. When selecting the protocol, matching is performed according to both name and number. We assigned a number to each protocol because we had previously experienced selection errors when matching only by name. At our facility, protocol names are written in Japanese in the RIS and in English on the CT console; the language difference makes name matching difficult. However, even if the names are written in English in both systems, matching by number would be beneficial due to better visibility. When there is a special note in the pre-check field regarding the scan range or scan phase, a card indicating the presence of the special note is placed on the CT console to prevent its application from being missed.

*4.6. Recording at the End of the Examination*

Although the radiation dose information on the CT scanner is transferred to a PACS server and is commercially available, dedicated dose management system, we mainly use RIS for routine radiation dose management due to its high flexibility. After the completion of imaging, the operator enters the total DLP for the entire examination into the RIS. When the head and trunk of a trauma patient are imaged in a single examination, the DLPs for the head and trunk scans are recorded separately. If there is planned or unplanned variance in the imaging procedures, its presence and content are also recorded in the RIS. Typical variances are shown in Table 3. Recording variance information at the time of imaging facilitates subsequent dose assessment.

**Table 3.** Variance from the imaging protocols.

| |
|---|
| Planned variance at pre-check |
| ■ Prolongation or shortening of the scan range |
| ■ Addition or omission of the scan phase |
| ■ Avoidance of exposure to the cardiac device |
| ■ Use of reduced contrast media dose and low voltage for renal hypofunction |
| Unplanned variance |
| ■ Body imaging without raising the arms |
| ■ Additional imaging due to patient motion |
| ■ Additional imaging due to poor breath holding |
| ■ Additional imaging to evaluate pathologies revealed by planned imaging |
| ■ Imaging the patient lying on their side |

## 5. Daily Radiation Dose Management

*5.1. Identification of Examinations to Be Investigated*

We monitor and analyze the radiation doses of all CT examinations every month. We collect information required for analysis, including the protocol number and name, total DLP for the entire examination, the presence of variance, age, sex, height, and weight, as a

comma-separated values (CSV) file. The Z-value of each examination, which represents the degree of the DLP deviation from the standard value for the patient weight, is automatically calculated using an original spreadsheet, and exceptionally high-dose examinations with large Z-values are extracted for investigation. Initially, examinations showing Z-values higher than 2 were investigated (Figure 2). However, because of the decline in the number of problematic cases, the threshold was elevated to a Z-value of 3 to improve efficiency. For examinations with variance, if the Z-value exceeds 2, the validity of the variance is examined, referring to clinical information and the purpose of the examination. When frequent, valid modifications of the scan range or imaging phase are noted, corresponding protocols may be created.

### 5.2. Causes of Exceptionally High Doses

Some potential causes of exceptionally high radiation doses are listed in Table 4. The DLP is high in heavy patients. However, high weight is not the cause of exceptionally high doses because the Z-value used to identify outliers is determined based on the relationship between weight and the DLP.

**Table 4.** Causes of exceptionally high radiation doses.

- Body imaging without raising the arms
- Planned or erroneous prolongation of the scan range
- Prolongation of the scan range to cover a huge mass lesion
- Planned or erroneous imaging at an additional phase
- Additional imaging due to patient motion
- Additional imaging due to poor breath holding
- Additional imaging to evaluate pathologies revealed by planned imaging
- Additional imaging due to failure of the CT scanner
- Selection of a protocol on the CT console different from that indicated in the RIS
- Off-center positioning of the patient
- Error in recording body weight
- Exceptionally large cross-section for weight

### 5.3. Problems with the Scan Range

Early after the start of radiation dose management practice, investigation of exceptionally high doses sometimes revealed excessive doses due to longer-than-necessary scan ranges. When attempting to identify the cause of inappropriate scan ranges, the definitions of the upper and lower margins of the range were found to be ambiguous and were made more specific. For example, the scan range for thoraco-abdominopelvic CT was changed from "including the lung apex and the inguinal region" to "from 2 cm above the lung apex to 1 cm below the inferior margin of the sciatic bone". In addition, since the verbal description of the scan range was misunderstood in some cases, a scan range list displaying the ranges on the localizer images was created to aid visual recognition. Moreover, training on setting up the scan range was reinforced for technologists inexperienced in CT practice. Presumably thanks to these efforts, outliers due to inappropriate scan range settings are rarely encountered at present.

### 5.4. Use of Water Equivalent Diameter

The radiation dose required to acquire diagnostic-quality images depends on the attenuation severity of the imaging section. The AEC system modulates the tube current and, consequently, radiation exposure by estimating the attenuation severity from the localizer images; thus, radiation exposure increases with increasing body size when using AEC. We use body weight as an indicator of body size for radiation dose management. However, even for the same weight, the size and attenuation severity of the imaging section may vary depending on body shape and tissue composition. The effective diameter and water equivalent diameter can be calculated as indicators of the section size using the dedicated dose management system. The effective diameter is the geometric mean of the anteropos-

terior and lateral dimensions of the cross-section, while the water equivalent diameter is calculated considering differences in X-ray attenuation severity between tissues [31]. When investigating an exceptionally high dose, checking the relationship between the water equivalent diameter and CTDIvol on the dose management system sometimes reveals that the cause of the high dose is a long water equivalent diameter and is not problematic.

### 5.5. Problems with Weight Records

In some cases, inaccuracies in weight records were inferred to be the cause of outliers. Formerly, although we measured body weight for enhanced CT, our weight records for unenhanced CT were mainly based on medical interviews. When investigating outliers, we encountered cases where the recorded weights were similar to those for previous examinations of the patients, despite apparent differences in body size observed on the acquired images, suggesting the possibility of the patients' unawareness of the weight change. Thus, in current practice, we measure the patient weight in the CT division just before the examination, regardless of the use of contrast media. Significant inaccuracy in weight records is no longer suspected in investigating outliers. For height, we usually rely on previous records in the electronic medical system or medical interviews because height is more constant than weight.

### 5.6. Limitation of the Analysis of DLP for Each Examination

We perform dose analysis mainly using the total DLP values per examination and variance records. Additional analysis is performed using the DLP and CTDIvol values for each series if necessary. With our analysis procedures, problems in a small proportion of the scan range are difficult to find. For example, we found abnormally strong radiation exposure at the top of the head in the CT component of PET/CT through slice-by-slice evaluation of tube currents [27,32]. Such strong exposure occurred when using the posteroanterior localizer image alone for AEC but not when using the posteroanterior and lateral images. Therefore, we decided to use two localizer images at our facility, while reducing the tube current for localizer imaging and informed the PET/CT vendor of the problem. Later, the AEC software was modified, and the abnormally strong exposure was resolved even when using the posteroanterior image alone [33]. For radiation dose management, it is recommended to examine tube current modulation along the slice location within a single imaging series. In addition, even if the radiation dose in a single examination is not high, the cumulative dose may become high due to frequent repetition of examinations. In our practice, we identify patients who received high cumulative doses over a 2-month period, provide the head of the relevant department with the information, and ask the head to review the validity of the referral with the members of the department. High cumulative doses often occur in severe, complicated patients for whom optimal planning of radiological imaging is not straightforward. We expect that the review of high cumulative doses may aid in the continuous improvement of the use of medical radiation.

### 5.7. Team Approach to Radiation Safety

At our facility, modality-specific quality assurance (QA)/quality control (QC) teams are organized to monitor and discuss radiological imaging services, and they hold monthly meetings. The results of CT dose analyses are reviewed in the CT QA/QC team meeting. In addition to radiation dose management matters, contrast media safety and other medical safety issues are discussed. The QA/QC team consists of radiologists, radiological technologists, and nurses. Nurses are not familiar with radiation safety issues, but they are familiar with general medical safety and have a viewpoint close to that of the patient; these characteristics should be beneficial to discussions of radiation safety. Regarding problematic cases, the facts, associated factors, root causes, and measures to prevent recurrence are discussed. Furthermore, comprehensive discussions are held on improving radiological imaging services, which may lead to the establishment of new imaging protocols, changes in imaging parameters, the development and revision of procedure manuals, and improvements in

the training of staff and communication with patients. Although staff members involved in radiological imaging services can make proposals for improving clinical practice to the responsible person at any time, periodical meetings should encourage constructive feedback and proposals from many staff members.

## 6. Summary

In this article, CT radiation dose management practice at our facility is presented. To optimize CT practice, imaging protocols are made visible in the protocol list and parameter list and are reviewed for continuous improvement. Clinical imaging is performed by selecting an optimal protocol for each examination. Radiation doses are analyzed for each protocol and each scanner and are reduced while preserving image quality and diagnostic performance. Problematic examinations are identified and discussed to improve CT imaging services. In our facility, many staff members involved in routine radiological imaging participate in radiation dose management activity. After the start of the activity, the number of problematic examinations decreased gradually. In addition to improving the documents and imaging service system, staff participation in dose management activities enhanced their awareness of radiation safety, presumably contributing to improving imaging practice. Radiation dose management is expected to play an important role in promoting radiation safety.

**Funding:** The author received no external funding in relation to this article.

**Institutional Review Board Statement:** Not applicable.

**Informed Consent Statement:** Not applicable.

**Data Availability Statement:** Not applicable.

**Acknowledgments:** The author thanks Hirofumi Hata, Kazunori Nagahara, and Hiroyasu Itoh for their support in preparing this manuscript and all of the staff involved in CT radiation dose management at Kitasato University Hospital.

**Conflicts of Interest:** The author declares no conflict of interest.

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
