# Peer review of "Radiation Dose Management in Computed Tomography: Introduction to the Practice at a Single Facility"

_tomography, doi:10.3390/tomography9030078_

Round 1

Reviewer 1 Report

This study reports a case of radiation dose management in CT examinations in a medical facility. This includes useful information for the comparison among facilities or for the reference of the CT management system in radiological protection. However, there are some unclear points. They should be adequately revised.

2. Management of Imaging Protocols

2.1. Significance of Protocol Management

It would be better to mention that DRLs are applied not for all patients but for standard-sized patients as well as in subsection 3.3.

Current relevant Japanese regulations recommend the use of DRLs for radiological protection for the patients. However, there is no mention of the actual use of DRL within the facility for CT dose management in this report.

2.3. Preparation of the Protocol List

It would be better to show not only the items of protocol but also concretely what kind of protocol and how many.

2.4. Tabulation of Imaging Parameters

The reasons to use word processor for the protocol list and spreadsheet for imaging parameters are not clear. The relationships between them and how to use them should also be described.

3. Overview of CT Radiation Dose Management

3.1. Dose Indices Used for Radiation Dose Management

Effective dose can indeed be calculated from DLP using conversion factors. However, it should be noted that there is a large uncertainty in its value. Does the author perform conversions of effective doses in actual CT dose management?

3.7. Dose Reference Tables for Referring Physicians

What does it mean by “standard effective dose values”? It is unclear.

The limitation of the use of CTDI and DLP have been noted so far, and SSDE has come to be used in recent due to the benefit to consider various body sizes of the patients. It would be better to mention it because water equivalent diameters and effective diameters are described in subsection 5.4.

4.2. Request for CT by Referring Physician

The doses of CT examinations vary so much depending the diagnoses and patients. How are the radiation exposures in CT examinations and effective dose values for natural radiation and certain radiological examination compared?

5.4. Use of Water Equivalent Diameter

In this paragraph, it is not clear whether the author has actually calculated the water equivalent diameters and effective diameters for checking the reason of high doses, as using the words “may reveal”.

Author Response

Comment 1)

  1. Management of Imaging Protocols

2.1. Significance of Protocol Management

It would be better to mention that DRLs are applied not for all patients but for standard-sized patients as well as in subsection 3.3.

Reply 1)

We added the comment.

Comment 2)

Current relevant Japanese regulations recommend the use of DRLs for radiological protection for the patients. However, there is no mention of the actual use of DRL within the facility for CT dose management in this report.

Reply 2)

We compare our doses to the DRLs in a usual manner, in addition to our original radiation dose management. We added the following description to the subsection “3.3. Need to Consider Body Size in Radiation Dose Management”.

“We determine median doses for standard-sized patients and compare them to the Japanese DRLs [20] every year and early after changing parameters.”

References)

  1. Kanda, R.; Akahane, M.; Koba, Y.; Chang, W.; Akahane, K.; Okuda, Y.; Hosono, M. Developing diagnostic reference levels in Japan. Jpn. J. Radiol. 2021, 39, 307–314. https://doi.10.1007/s11604-020-01066-5.

Comment 3) 

2.3. Preparation of the Protocol List

It would be better to show not only the items of protocol but also concretely what kind of protocol and how many.

Reply 3)

As indicated in the subsection “2.2. Preparation of Subdivided Imaging Protocols”, we have many protocols. Our main imaging protocol list has 108 pages (A4, 10 points), and, additonally, there are lists for the emergency department and operation department. We intend to provide an overview of CT radiation dose management, and detailed explanation of the protocols is beyond the scope. The kind and number of required protocols differ among facilities, and thus we described the concept of subdivision of the protocols.

Comment 4)

2.4. Tabulation of Imaging Parameters

The reasons to use word processor for the protocol list and spreadsheet for imaging parameters are not clear. The relationships between them and how to use them should also be described.

Reply 4)

The content in the protocol list is variable depending on the protocol, and thus tabulation is not suitable for the protocol list. For example, some multiphase imaging protocols require lengthy explanation of the scan range for each phase. Explanation for contrast infusion is also lengthy especially when using bolus tracking, whereas it is not needed for plain CT.

In contrast, the imaging parameters are generally simple. They are described as a number, a single word, or a few words. Tabulation is useful for visibility.

The imaging protocol list is the main list, and is supplemented by the imaging parameter list. To clarify the relation between the two lists, we revised the first two sentences of the subsection "2.4. Tabulation of Imaging Parameters” as follows:

“To supplement the imaging protocol list, we have prepared the imaging parameter list using general-purpose spreadsheet software. Detailed parameters of each imaging protocol for image acquisition and reconstruction are described in the imaging parameter list”

Comment 5)

  1. Overview of CT Radiation Dose Management

3.1. Dose Indices Used for Radiation Dose Management

Effective dose can indeed be calculated from DLP using conversion factors. However, it should be noted that there is a large uncertainty in its value. Does the author perform conversions of effective doses in actual CT dose management?

Reply 5)

We provide effective dose estimates for referring physicians and patients to facilitate their understanding of the radiation dose level. However, we use DLP and CTDIvol for radiation dose management in the radiology department. In the subsection, we compare the total DLP, DLP in a series, and CTDIvol, and explain the importance of total DLP because of its relationship with effective dose.

For clarification, we added the following comment:

“the total DLP in an examination is considered to represent the potential detriment of an examination. Thus,”

Comment 6)

3.7. Dose Reference Tables for Referring Physicians

What does it mean by “standard effective dose values”? It is unclear.

Reply 6)

The word “standard” corresponds to “standard” in “standard DLP”. We added the following sentence to the last of the first paragraph of the subsection:

“The standard effective dose represents effective dose estimated at 60 kg for the respective order item.”

Comment 7)

The limitation of the use of CTDI and DLP have been noted so far, and SSDE has come to be used in recent due to the benefit to consider various body sizes of the patients. It would be better to mention it because water equivalent diameters and effective diameters are described in subsection 5.4.

Reply 7)

We added the following description to the subsection “3.1. Dose Indices Used for Radiation Dose Management”:

“The absorbed dose varies depending on the section size, even when radiation output from the CT scanner is identical. Size-specific dose estimate (SSDE) is a relatively new index of CT radiation dose and is calculated considering the effect of the section size [10,11]. Whereas CTDIvol represents radiation output from the scanner, SSDE is considered a better indicator of the patient’s absorbed dose than CTDIvol. However, the estimation of SSDE requires the section size, and a CT scanner does not automatically provide the SSDE value. CTDIvol and DLP are still typically used for CT radiation dose management.”

References)

  1. American Association of Physicists in Medicine (AAPM). Size-specific dose estimates (SSDE) in pediatric and adult body CT examinations (Task Group 204); American Association of Physicists in Medicine: College Park, MD, USA, 2011.
  2. American Association of Physicists in Medicine (AAPM). Size specific dose estimate (SSDE) for head CT (Task Group 293); American Association of Physicists in Medicine: College Park, MD, USA, 2019.

Comment 8)

4.2. Request for CT by Referring Physician

The doses of CT examinations vary so much depending the diagnoses and patients. How are the radiation exposures in CT examinations and effective dose values for natural radiation and certain radiological examination compared?

Reply 8)

I agree about the difficulty of comparison among radiation exposures from different sources. As recommended generally, we propose the use of effective dose in our facility. In the document for explanation to the patient, we describe general explanation of radiation exposure in CT and effective dose values for natural radiation and certain radiological examinations. Effective dose estimates for each order item are provided in the dose reference tables, which is accessible on the electronic ordering system. We have many order items, despite much less than imaging protocols, and dose differences depending on the diagnosis are roughly reflected by those depending on the order item. At the top of the dose reference table, the higher dose in a larger patient is commented. The dose reference tables are prepared not only for CT but also for plain radiography, fluoroscopy, angiography, and radionuclide imaging.

We revised the last sentence of the subsection as follows:

“The referring physician explains the dose, justification, and optimization of the individual examination with reference to this document and the dose reference table presenting a standard effective dose value for each order item.”

Comment 9)

5.4. Use of Water Equivalent Diameter

In this paragraph, it is not clear whether the author has actually calculated the water equivalent diameters and effective diameters for checking the reason of high doses, as using the words “may reveal”.

Reply 9)

We actually use water equivalent diameters in investigating an exceptionally high dose. We replaced “may reveal” by “sometimes reveal”.

Reviewer 2 Report

Abstract

It is stated that the dose received by patients in CT is also high. Since the magnitude of CT dose should be related to other dose levels, I suggest to change high dose in relatively high dose.

The dose should not be minimum to obtain diagnostic-quality images, but a low as reasonably achievable (to obtain diagnostic-quality images)

Many staff members are involved in dose management. However, it is unclear how much time is spend on the dose management in this hospital. Please provide data on time spend.

Introduction

Please again, change high dose in relatively high dose.

No medical physicist are involved in this dose management. It is not argued in the paper why involvement of medical physicist is needed, wanted or at least mandatory to obtain a good radiation dose management system. See e.g. McCollough AJR 2016 206:6; 1241. Caruana Physica Medica 48 (2018) 162-168.

Management of Imaging Protocols

Radiation dose in CT is not only determined by imaging protocol and patient size, but also by the specific CT scanner used. Please add.

If the radiation dose used in a facility is higher than the DRL, it triggers inspection for dose reduction. Should read: should trigger inspection.

We evaluate radiation dose and image quality for each protocol and monitor the frequency of utilization of each protocol and validity of the protocol selection for each examination. It is not clear how and when the radiation dose and image quality of each protocol is monitored.

For children, we use a low-dose protocol at the expense of increased image noise, considering high radiosensitivity and long life expectancy of children. Why is a low-dose protocol used for children and decreased image quality? Especially for children you want good image quality with low noise (at ALARA dose) in order to be able to exclude pathology.

We have prepared a protocol list using general-purpose word-processing software to facilitate appropriate use and review of the imaging protocols. What does general-purpose word-processing software mean? E.g. MS word?

First, a revised version of the imaging parameter list is proposed, and the responsible person and other CT team members confirm the validity of the revision. Who is responsible for the imaging protocols? Who is involved in revision?

Overview of CT Radiation Dose Management

Automatic Exposure Control. Modern CT systems also use (semi) automatic modulation of the tube voltage based on patient size in the topogram. Please add adaption of kVp in this section.

Maximum and minimum tube currents may be set in AEC software. Why would you set tube current limits in the AEC software? Would it not be much better to set the tube voltage for certain protocols? In modern CT scanners with tube voltage selection based on body size, the tube voltage is often set lower with an increase tube current.

Radiation Dose Management Considering Body Size. A method for analysing DLP against weight is presented. However, example data is missing. Please provide example data for the dose management method presented.

Standard Dose Tables for Radiation Dose Management. It is unclear how potential problems are indicated from the standard dose tables.

Identifying Examinations Delivering Exceptionally High Doses. When analyzing DLP-weight plots to obtain standard dose determination equations, we calculate DLP deviations, defined as actual DLP values provided by the scanner minus DLP values estimated based on the weight and equation, and determine their standard deviations (SDs). It is unclear what is meant by DLP values estimated based on the weight and equation. Please provide example data from your clinical practice.

Dose Reference Tables for Referring Physicians. The standard dose tables are used for radiation dose management in the radiology department. We also prepare dose reference tables for referring physicians and make them accessible via the electronic medical system terminal used to order examinations. Please show these tables as you use them in clinical practice.

When the referring physician selects an order item, a pop-up window appears presenting the following text: “Please consider the indication, range, and frequency, taking account of the possibility of radiation-induced cancer.” This is a very way to make the requestion physician aware of dose levels for the patient, and justification of the examination for this specific patient! Do you have information on dose levels before and after implementation of this feature? Does it really contribute to better justification and dose reduction?

Recording at the End of the Examination. Modern CT scanners also provide the option of dose notifications. That is, pop-ups at the CT console when certain dose levels are exceeded. This has not been implemented in your facility? Why not?

Daily Radiation Dose Management

Identification of Examinations to be Investigated. The proposed method seems to be fully build on manual tools in Excel. Since there are many dose management software tools on the market which can do this task, why do the authors not advocate the use of this software? One of the clear advantages is that this software provides instant information on dose levels.

Use of Water Equivalent Diameter. How is the effective diameter of patients determined? Manually? Do the authors have Size-Specific Dose Estimates (SSDE) implemented?

Problems with Weight Records. Is the weight of all patients standard measured? And the height of patients?

Limitation of the Analysis of DLP for Each Examination. In our practice, patients who received high cumulative doses over a 2 month period are monitored and the responsible person of the relevant department is notified. How do you identify these patients? What is considered a high cumulative dose? And what is meant by the responsible person? The referring physician? Since each referral has to be justified, how can a referring physician include this information in new referrals?

Author Response

Comment 1)

It is stated that the dose received by patients in CT is also high. Since the magnitude of CT dose should be related to other dose levels, I suggest to change high dose in relatively high dose.

Reply 1)

We revised the Abstract according to the reviewer’s comment.

Comment 2)

The dose should not be minimum to obtain diagnostic-quality images, but a low as reasonably achievable (to obtain diagnostic-quality images)

Reply 2)

We replaced “the minimum necessary irradiation” by “appropriate adjustment of irradiation” in the subsection “3.2. Automatic Exposure Control”.

We also revised the Abstract (line 14) as follows:

Comment 3)

Many staff members are involved in dose management. However, it is unclear how much time is spend on the dose management in this hospital. Please provide data on time spend.

Reply 3)

Unfortunately, we do not have data on time spent.

Comment 4)

Introduction

Please again, change high dose in relatively high dose.

Reply 4)

We revised the Introduction according to the reviewer’s comment.

Comment 5)

No medical physicist are involved in this dose management. It is not argued in the paper why involvement of medical physicist is needed, wanted or at least mandatory to obtain a good radiation dose management system. See e.g. McCollough AJR 2016 206:6; 1241. Caruana Physica Medica 48 (2018) 162-168.

Reply 5)

We added the following comment:

“although the critical roles of medical physicists are undisputed”

and cited the papers indicated by the reviewer.

  1. McCollough, C.H. The role of the medical physicist in managing radiation dose and communicating risk in CT. AJR Am. J. Roentgenol. 2016, 206, 1241–1244. https://doi.10.2214/AJR.15.15651.
  2. Caruana, C.J.; Tsapaki, V.; Damilakis, J.; Brambilla, M.; Martín, G.M.; Dimov, A.; Bosmans, H.; Egan, G.; Bacher, K.; McClean, B. EFOMP policy statement 16: The role and competences of medical physicists and medical physics experts under 2013/59/EURATOM. Phys. Med. 2018, 48, 162–168. https://doi.10.1016/j.ejmp.2018.03.001.

Comment 6)

Management of Imaging Protocols

Radiation dose in CT is not only determined by imaging protocol and patient size, but also by the specific CT scanner used. Please add.

Reply 6)

We added “CT scanner” in the subsection “2.1. Significance of Protocol Management”.

Comment 7)

If the radiation dose used in a facility is higher than the DRL, it triggers inspection for dose reduction. Should read: should trigger inspection.

Reply 7)

We replaced “triggers” by “should trigger” in the subsection “2.1. Significance of Protocol Management”.

Comment 8)

We evaluate radiation dose and image quality for each protocol and monitor the frequency of utilization of each protocol and validity of the protocol selection for each examination. It is not clear how and when the radiation dose and image quality of each protocol is monitored.

Reply 8)

The frequency and timing of these processes are variable, and the following is our general policy.

We evaluate radiation dose and image quality for each protocol early after the introduction and parameter change. Comparison with the DRLs is performed annually for the protocols for which national DRLs are established. The dose reference tables are revised every year. Standard dose tables are revised every six months until 200 examinations are pooled. The frequency of utilization of each protocol is assessed every year. The protocol selection is made at the precheck, and its validity is assessed by the operator at the examination and by the radiologist at the image interpretation. When a problem regarding protocol selection is noticed, similar problems are searched.

Comment 9)

For children, we use a low-dose protocol at the expense of increased image noise, considering high radiosensitivity and long life expectancy of children. Why is a low-dose protocol used for children and decreased image quality? Especially for children you want good image quality with low noise (at ALARA dose) in order to be able to exclude pathology.

Reply 9)

I recognize that the description was misleading. It was revised as follows:

“For children, we reduce radiation dose with special care while maintaining clinically acceptable image quality,”

Comment 10)

We have prepared a protocol list using general-purpose word-processing software to facilitate appropriate use and review of the imaging protocols. What does general-purpose word-processing software mean? E.g. MS word?

Reply 10)

We use Microsoft WORD.

Comment 11)

First, a revised version of the imaging parameter list is proposed, and the responsible person and other CT team members confirm the validity of the revision. Who is responsible for the imaging protocols? Who is involved in revision?

Reply 11)

The head of the radiology department (the author, radiologist) is responsible for the imaging protocol. Radiologists and radiological technologists are involved in revision. I suppose that this assignment is not generalizable because the head of the radiology department is not necessarily familiar with the physical and technical aspects of radiological imaging.

Comment 12)

Overview of CT Radiation Dose Management

Automatic Exposure Control. Modern CT systems also use (semi) automatic modulation of the tube voltage based on patient size in the topogram. Please add adaption of kVp in this section.

Reply 12)

We added the following description to the subsection “3.2. Automatic Exposure Control”:

“In addition, automatic tube voltage selection techniques aid the selection of appropriate tube voltage for each imaging series to reduce radiation dose according to the patient size and the type of the examination (plain CT, contrast-enhanced CT, or CT angiography) [18,19].”

References)

  1. Yu, L.; Li H.; Fletcher, J.G.; McCollough, C.H. Automatic selection of tube potential for radiation dose reduction in CT: a general strategy. Med. Phys. 2010, 37, 234–243. https://doi.10.1118/1.3264614.
  2. Frellesen, C.; Stock, W.; Kerl, J.M.; Lehnert, T.; Wichmann, J.L.; Nau, C.; Geiger, E.; Wutzler, S.; Beeres, M.; Schulz, B.; et al. Topogram-based automated selection of the tube potential and current in thoraco-abdominal trauma CT - a comparison to fixed kV with mAs modulation alone. Eur. Radiol. 2014, 24, 1725–1734. https://doi.10.1007/s00330-014-3197-7.

Comment 13)

Maximum and minimum tube currents may be set in AEC software. Why would you set tube current limits in the AEC software? Would it not be much better to set the tube voltage for certain protocols? In modern CT scanners with tube voltage selection based on body size, the tube voltage is often set lower with an increase tube current.

Reply 13)

We have CT scanners from Siemens, GE Healthcare, Fujifilm Healthcare, and Canon Medical Systems. Maximum mA and minimum mA are parameters to be set for the scanners other than the Siemens scanner. Changes in tube current are more pronounced for the scanners than for the Siemens scanner operated with average/average modulation, which may be related to the differences in parameters to be set.

Comment 14)

Radiation Dose Management Considering Body Size. A method for analysing DLP against weight is presented. However, example data is missing. Please provide example data for the dose management method presented.

Reply 14)

We added Figure 1 explaining the method for analyzing the DLP-weight relationship.

Comment 15)

Standard Dose Tables for Radiation Dose Management. It is unclear how potential problems are indicated from the standard dose tables.

Reply 15)

Using the standard dose tables, we compare radiation doses between CT scanners and imaging protocols to identify unreasonable inconsistencies, as stated in the text. For example, when a difference between scanners for a given imaging protocol is apparent, its validity is investigated. On the other hand, a difference between imaging protocols using a given scanner may be mismatched with expectation.

Comment 16)

Identifying Examinations Delivering Exceptionally High Doses. When analyzing DLP-weight plots to obtain standard dose determination equations, we calculate DLP deviations, defined as actual DLP values provided by the scanner minus DLP values estimated based on the weight and equation, and determine their standard deviations (SDs). It is unclear what is meant by DLP values estimated based on the weight and equation. Please provide example data from your clinical practice.

Reply 16)

We added Figure 2 explaining the identification of examinations delivering exceptionally high doses.

Comment 17)

Dose Reference Tables for Referring Physicians. The standard dose tables are used for radiation dose management in the radiology department. We also prepare dose reference tables for referring physicians and make them accessible via the electronic medical system terminal used to order examinations. Please show these tables as you use them in clinical practice.

Reply 17)

In this communication, we intend to overview our radiation dose management strategy.

Comment 18)

When the referring physician selects an order item, a pop-up window appears presenting the following text: “Please consider the indication, range, and frequency, taking account of the possibility of radiation-induced cancer.” This is a very way to make the requestion physician aware of dose levels for the patient, and justification of the examination for this specific patient! Do you have information on dose levels before and after implementation of this feature? Does it really contribute to better justification and dose reduction?

Reply 18)

We do not have information about the effect of the pop-up window on the referral.

Comment 19)

Recording at the End of the Examination. Modern CT scanners also provide the option of dose notifications. That is, pop-ups at the CT console when certain dose levels are exceeded. This has not been implemented in your facility? Why not?

Reply 19)

As recommended generally, we set a notification value for each protocol, and the operator inputs the reason when a message window appears. However, the reason is “a large patient” in most cases because we prepare subdivided imaging protocols and the operator does not customize the parameters for each examination. We record variance information in the RIS, regardless of whether the notification value is exceeded or not, and use the information in routine monitoring.

Comment 20)

Daily Radiation Dose Management

Identification of Examinations to be Investigated. The proposed method seems to be fully build on manual tools in Excel. Since there are many dose management software tools on the market which can do this task, why do the authors not advocate the use of this software? One of the clear advantages is that this software provides instant information on dose levels.

Reply 20)

Although we send radiation dose information to a commercial dose management system, we mainly use RIS for routine radiation dose management due to its high flexibility, as stated at the top of the subsection “4.6. Recording at the End of the Examination”. We added “commercially available” to the statement.

I proposed incorporating a Z-value-based function into the dose management system for the vendor; however, it is not realized. We mainly use the system to calculate water equivalent diameters in our routine CT dose management.

Comment 21)

Use of Water Equivalent Diameter. How is the effective diameter of patients determined? Manually? Do the authors have Size-Specific Dose Estimates (SSDE) implemented?

Reply 21)

We calculate water equivalent diameters using the commercial dose management system. The word “commercial” was added.

As for SSDE, we sometimes investigate SSDEs obtained by the system in body CT and those calculated from CTDIvol and water equivalent diameters in brain CT. SSDE calculation in brain CT is not incorporated into the system.

We added the following description to the subsection “3.1. Dose Indices Used for Radiation Dose Management”:

"The absorbed dose varies depending on the section size, even when radiation output from the CT scanner is identical. Size-specific dose estimate (SSDE) is a relatively new index of CT radiation dose and is calculated considering the effect of the section size [10,11]. Whereas CTDIvol represents radiation output from the scanner, SSDE is considered a better indicator of the patient’s absorbed dose than CTDIvol. However, the estimation of SSDE requires the section size, and a CT scanner does not automatically provide the SSDE value. CTDIvol and DLP are still typically used for CT radiation dose management.”

References)

American Association of Physicists in Medicine (AAPM). Size-specific dose estimates (SSDE) in pediatric and adult body CT examinations (Task Group 204); American Association of Physicists in Medicine: College Park, MD, USA, 2011.

American Association of Physicists in Medicine (AAPM). Size specific dose estimate (SSDE) for head CT (Task Group 293); American Association of Physicists in Medicine: College Park, MD, USA, 2019.

Comment 22)

Problems with Weight Records. Is the weight of all patients standard measured? And the height of patients?

Reply 22)

We currently measure the patient weight in the CT division just before the examination, as described in the subsection.

For height, we usually rely on previous records in the electronic medical system or medical interviews because height is more constant than weight. This is added to the subsection.

Comment 23)

Limitation of the Analysis of DLP for Each Examination. In our practice, patients who received high cumulative doses over a 2 month period are monitored and the responsible person of the relevant department is notified. How do you identify these patients? What is considered a high cumulative dose? And what is meant by the responsible person? The referring physician? Since each referral has to be justified, how can a referring physician include this information in new referrals?

Reply 23)

We extract DLP values in two months from the RIS, convert DLPs to effective doses, and calculate the cumulative effective dose in each patient. The definition of the high cumulative dose has not been fixed. The information about high cumulative doses is given to the responsible person of the relevant department, i.e., the head of the department. Usually, two or more doctors ordered CT for the patient of interest. High cumulative doses are usually encountered in severe, complicated patients in whom justification and optimization are not necessarily simple. We think that discussion in the department about high cumulative doses will be useful for future practice.

We revised the last sentence of the subsection “5.6. Limitation of the Analysis of DLP for Each Examination” as follows:

“In our practice, we identify patients who received high cumulative doses over a 2-month period, provide the head of the relevant department with the information, and ask the head to review the validity of the referral with the members of the department. High cumulative doses often occur in serious, complicated patients for whom optimal planning of radiological imaging is not straight forward. We expect that review of high cumulative doses may aid continuous improvement of use of medical radiation.”

Round 2

Reviewer 2 Report

Thank you for this revision. All questions nicely answered and improvemets in the manuscipt made. No further comments.